# Promises and Presuppositions of Biomimicry

**DOI:** 10.3390/biomimetics5030033

**Published:** 2020-07-09

**Authors:** Rebecca Barbara MacKinnon, Jeroen Oomen, Maibritt Pedersen Zari

**Affiliations:** 1Graduate School of Life Sciences, Utrecht University, Padualaan 8, 3584 CH Utrecht, The Netherlands; 2Urban Futures Studio, Utrecht University, Heidelberglaan 8, 3584 CH Utrecht, The Netherlands; j.j.oomen@uu.nl; 3Wellington School of Architecture, Victoria University of Wellington, Wellington 6012, New Zealand; maibritt.pedersen@vuw.ac.nz

**Keywords:** biomimicry, promise, sustainability, innovation

## Abstract

Under the umbrella of biologically informed disciplines, biomimicry is a design methodology that proponents often assert will lead to a more sustainable future. In realizing that future, it becomes necessary to discern specifically what biomimicry’s “promises” are in relation to sustainable futures, and what is required in order for them to be fulfilled. This paper presents research examining the webpages of the Biomimicry Global Network (BGN) to extract the claims and promises expressed by biomimicry practitioners. These promises are assessed using current literature to determine their presuppositions and requirements. Biomimicry’s promises are expressed in terms of potential for innovation, sustainability, and transformation and appear to depend on perceived relationships between humanity and nature; nature and technology; the underlying value judgements of practitioners. The findings emphasize that in order for the communicated promise of biomimicry to be realized, a particular ethos and respectful engagement with nature must accompany the technological endeavors of the practice.

## 1. Introduction

The flow of biological knowledge into the field of design has led to novel research across many disciplines including materials science, architecture, urban design, computer science, and robotics [1,2]. Within this arena of innovation, biomimicry is a distinct discipline [3,4]. Although it has a longer history, biomimicry was popularized after Janine Benyus published *Biomimicry: Innovation Inspired by* Nature in 1997 [5]. The seminal publication consolidated the concept of biomimicry, differentiating it from practices developing in parallel by emphasizing its focus on sustainability. Since then, the expressed potential of biomimicry has increased rapidly, demonstrated by the increasing number of associated practitioners, educational programs, and patented projects [6].

According to advocates, technological biomimicry is brimming with promise. Mathews [7], for example, argues that a second industrial revolution based on biomimicry is pending and that it is a revolution that could change the world beyond recognition. Biomimicry has been described as a field that fascinates [6]; that presents novel ways of viewing and valuing nature [8]; that has an approach that will “inspire new mindsets, values and narratives concerning the relationship between people and nature and alternative visions of development” [9]. In the words of Biomimicry South Africa, biomimicry is “a game changer: after we hear it and understand it, we never see the world quite the same ever again (….) It’s one of the most inspiring approaches to the world’s big challenges that you’ll discover”.

Taking inspiration from nature is not new to innovation, nor is it unique to biomimicry. The term “biomimetic promise” was coined by von Gleich et al. [6] to capture the promise of bio-inspired designs (including biomimetics, bionics, and biomimicry). It suggests that, by taking inspiration from nature, and thus relying on evolutionary optimization, bio-inspired solutions ought to be innovative, but also ecologically sound, resilient, and low risk [6,9]. This promise links bio-inspired fields to the discourse on sustainability [9]. However, although biomimetics, bionics, and biomimicry are concerned with the transfer of knowledge from the natural to the human-made world, they rest on different intentions, levels of mimicry, and are often associated with disparate disciplines [3,10]. The fulfilment of the biomimetic promise may thus differ for each practice. Within this paper, the term biomimetic is used as an adjective to refer to solutions associated with biomimicry, and not to the practice of biomimetics.

Advocates of biomimicry have promised that it will spark a technological, environmental, and even social revolution [11]. Beyond the delivery of sustainable design, biomimicry is presented as a new lens through which to view the natural world [5,12]. By promoting a shift in perspectives, hierarchies, and beliefs, biomimicry is, according to its advocates, working towards recognizing the intrinsic value of nature [13]. It is based on a view of nature as a source of knowledge rather than an object of knowledge [12], and natural systems as inspiration rather than as resource. What biomimicry practitioners and advocates seem to promise is the disintegration of the idea of the human as having domain over nature as it is now known, thereby eliminating the idealized image of humans as a supreme species [11]. Benyus refers to this as an unsettling of human exceptionalism and as biomimicry’s most profound contribution [5]. By exploring nature for inspiration, the hope is that as observers, humanity will engender a more respectful, responsible, and humble engagement with not only nonhuman, but human life as well [11].

Expressing these high hopes, practitioners make technological promises and create expectations around the effectiveness of biomimicry as a means to achieve sustainable and transformative change. Biomimicry’s promises are issued as hopeful enactments of a desired future. These promises matter, because they are an attempt to raise both general enthusiasm and specific expectations around biomimicry. In describing such promises, they may become widely shared expectations of potential future technological situations that attract investment [14]. Promises and expectations matter, because they provide a “prospective structure” that shapes practitioners, policymakers, and potential investors’ agendas and actions [15,16,17,18]. Firms and policymakers are often confronted with technological expectations and promises upon which they have to make decisions. Many of those reveal disappointing outcomes [14]. Not only do promises structure decision-making in the present, they also co-produce an imaginary about what a desirable future ought to be like. Such socio-technical imaginaries, collectively held, institutionally stabilized, and publicly performed visions of desirable futures, animated by shared understandings of forms of social life, and social order attainable through, and supportive of, advances in science and technology, further come to shape technological development, impacting the kind of future worlds that are possible [19]. This means is important to develop analytical perspectives to make sense of the promissory and future-oriented properties of biomimetic innovations.

Much of the public engagement with biomimicry stems from the Biomimicry Institute in Montana, USA. Established in 2006, the Institute’s mission is to “naturalize culture by promoting the transfer of ideas, designs, and strategies from biology to sustainable human systems design” [4]. The Biomimicry Institute has grown to mobilize tens of thousands of biomimicry practitioners, or biomimics, in what is known as the Biomimicry Global Network, or BGN. This network of practitioners plays an important role in creating a narrative around biomimicry and its promises. As much of the research on biomimicry has been theoretical or philosophical in nature (see: [7,11,20,21,22,23]), an investigation into the imaginary of biomimicry as proposed by the BGN and its affiliates is a necessary and worthwhile endeavor.

This article examines the promises expressed by affiliates of the Biomimicry Global Network in relation to innovation and sustainability, and the basis upon which these promises are made. The categories of innovation, sustainability and transformation form the crux of the investigation, linking to the intentions for practicing biomimicry as described by Pedersen Zari [24], namely biomimicry for innovation; biomimicry for sustainability; biomimicry for wellbeing. The BGN is arguably biomimicry’s most influential and charismatic representation and, as such, the one with the most power to shape the imaginary of biomimicry. The promises issued by this body of practitioners are analyzed using the texts of their publicly available webpages. By investigating each network’s public message, the collective narrative of the BGN can be understood and critically analyzed.

Given the technological promises issued by its practitioners, it is important to investigate biomimicry’s promises carefully. Analyzing these grounds upon which sustainability claims are made is critical to be able to assess the merits of biomimicry promises. For practicing or aspiring biomimics, this analysis encourages a critical and reflexive view of the practice; revealing its potential pitfalls, nuances, and presuppositions. Such an investigation of biomimicry adds an important reminder of the preconditions for the realization of a promising practice to biomimicry literature. This study is a comparative analysis making use of accessible public domains to investigate the promises of biomimicry and to then contextualize them within relevant academic literature. The study represents a critical reflection of the field, aiming to highlight the types of promises and expectations present in biomimicry.

## 2. Materials and Methods

This study utilized a number of text mining techniques to investigate the promises of biomimicry as they are communicated by the Biomimicry Global Network (BGN). The process involved data retrieval, extraction, and review. The underlying objective of utilizing these sources was to understand how biomimicry is communicated by BGN practitioners, and how such promises appear to the general public. The text analysis gives a cursory overview of what biomimicry claims to offer. These proclaimed promises were assessed using publications from the fields of philosophy, social science, environmental anthropology, sustainability, and design.

### 2.1. Data Retrieval

The Biomimicry Global Network is an affiliation of practitioners and organizations spread across 21 countries, 36 regional networks, and encompasses 12,576 members. The network stems from the Biomimicry Institute in Montana, USA. The Institute’s website maps and links each webpage of the global network. This network map acted as the database for data retrieval. The webpages used in this study were selected from this map based on their accessibility, usability, and language. Precisely, this means that the selected websites were accessible via a functioning weblink; established as an authorized website (thus excluding Facebook pages and blogs); contained useable content such as “what is biomimicry?”, “biomimicry 101” or similar sections; finally, either had a built-in English translation function, or could be translated using Google Translate.

### 2.2. Data Extraction

Upon visiting each webpage, the appropriate text was extracted. The main objective of the evaluation was to understand how the idea of biomimicry was introduced; thus, information was collected from “what is biomimicry?” or equivalent content from each webpage. Within these sections, each body of text was copied over to a separate Word document. Using the web-based reading and analysis environment, Voyant Tools (v. 2.4), each separate document was uploaded, and the texts could be analyzed as a combined corpus.

### 2.3. Data Review

The Voyant Tools interface consists of five boxes, each displaying a different mining or visualization tool. The layout is customizable and interactive, meaning each box can be updated to isolate specific terms, contexts, or documents within the corpus. The standard display includes a cirrus cloud of frequently used words; a text reader; text trends; a summary of the entire corpus; as well as contextual phrases and associations.

The summary, reader, and context boxes were the primary tools used to analyze the BGN texts. The summary tool provided an overview of the text corpus. It included categories such as document length, vocabulary density, as well as frequently used or distinctive words. The context box was used to analyze the potential overlap embedded in the combined corpus. Repetition of phrases, definitions, and references to either Janine Benyus or the Biomimicry Institute were noted and assessed.

The reader and context boxes were used to create a word tree of phrases associated with biomimicry’s communicated promise. To create the word tree, verbs that followed “biomimicry” in the corpus were highlighted and used as keywords. For example, by using the word “offer” as a keyword, but inputting it as “offer*”, all variations of the word (offer, offers, offering) could be seen within the contexts they arose throughout the corpus. The associated phrases were used to fill the last branch of the word tree. The phrases were quoted directly from the corpus within 10 words. The text analyses acted as a basis for analyzing the BGN narrative and further understanding the promises of biomimicry.

## 3. Results

### 3.1. Biomimicry Communicated

A total of 19 webpages were retrieved from the Biomimicry Global Network (BGN) and utilized in the analysis. Table 1 lists the names, web addresses, and content of each network. The content refers to either a distinct tab visible from the homepage, or the pathway to the relevant content page. The body of text from these content pages was extracted and added to the text corpus.

The complete corpus consisted of 5735 words, of which 1298 were unique word forms (those other than articles and highly repeated words). The overlap across the texts within the corpus suggests that the BGN has an overall branded messaging and collective narrative. Of the 19 webpages analyzed, ten utilize similar or identical definitions; eight reference Janine Benyus or the Biomimicry Institute; four mention Benyus’ book “Biomimicry: Innovation Inspired by Nature” [5]. After “biomimicry’, the most commonly used words were:nature (73)design (53)life (42)solutions (38)sustainable (35)

The word tree in Figure 1 displays phrases used in association with biomimicry’s aspirations. In order of frequency, the tree displays verbs and corresponding phrases that followed the word “biomimicry” in the corpus. The dots shown alongside indicate how each phrase relates to the potential of biomimicry for innovation, sustainability and transformation. As previously mentioned, these link to the three main intentions for pursuing biomimicry [24]. Where more than one dot is shown, the most applicable color is indicated first.

### 3.2. Understanding Biomimicry’s Promise

The text analysis revealed the clustered vision and promissory narrative created by the Biomimicry Global Network. By applying future-oriented terms, as is shown in the word tree, biomimicry supposes a more desirable future and in so doing presupposes an undesirable present; which is unsustainable, non-regenerative, ill-adapted, and even unhealthy. Consequently, biomimicry’s promise ties into a widely shared desire for a transition from our current reality to a potentially more sustainable future. The collective BGN narrative expresses promise in the realms of innovation, sustainability, and transformation. These categories link to the intentions for practicing biomimicry as described by Pedersen Zari [24]. The categories are further utilized to understand biomimicry’s promise as it is communicated by the BGN.

#### 3.2.1. Innovation

“Biomimicry is an approach to innovation that seeks sustainable solutions to human challenges by emulating nature’s time-tested strategies.” This definition appears on the Biomimicry Institute’s webpage and is quoted on another three affiliate websites. The definition lays out the fundamentals of what biomimicry is, what it aims to accomplish, and how. The *how* of this definition refers to the epistemology of biomimicry, namely: biomimetic innovation relies on the emulation (which involves discovery and translation) of nature’s time-tested (or evolutionarily optimized) strategies. This approach is emphasized throughout the text corpus with the repetition of words such as “emulate” (28) or “learn” (33), and notably by the many phrases used to describe nature’s time-tested strategies. These phrases include terms such as “genius” (13), “consummate engineers” (8), or “3.8 billion years” (11)—referring to the evolutionary time scale of natural phenomena. Furthermore, mentions of nature as having untapped wisdom, being a catalog of products, or as a library of efficient designs all link to the basis of biomimetic innovation and to the biomimetic promise. Only three of the 19 webpages separate biomimicry from its often-synonymous contemporaries, biomimetics and bionics. Throughout the corpus, and as displayed in the word tree, biomimicry’s potential for innovation is communicated as a new way of looking at design, entrepreneurship, and business.

#### 3.2.2. Sustainability

Sustainability is the most prominently expressed claim throughout the texts of the BGN webpages, with variations of the word repeated 52 times. Biomimicry is proposed as a pathway to sustainable designs; as well as a positive way of communicating and encouraging sustainability. It is clear from the text corpus that, for BGN affiliates, the concept of bio-inspiration is pursued in an attempt to actively achieve sustainability. The goal of biomimicry is repeated four times, stating that it aims to “create products, processes, and policies—new ways of living—that are well-adapted to life on Earth over the long haul”. In order of frequency, the adjectives used to describe biomimetic designs are, “sustainable” (35), “well-adapted” (6), “life-friendly” (5), “innovative” (5), and “efficient” (4). One website (Biomimicry Iberia) goes on to elaborate that the sustainability goals are tested against biomimicry’s characteristic Life Principles and are in that way assessed for their appropriateness.

In delivering increasing requirement for sustainability (as noted in the word tree), biomimicry relies on mimicry at three levels: form, process and ecosystem. The key difference in biomimicry motivated for innovation and biomimicry for sustainability is mimicking not only organisms, but also the underlying processes and functions of ecosystems [24]. Mimesis as a multi-level approach, mimicking forms and processes within the context of their ecosystems, or simply emulating ecosystems themselves, is said to be the most likely to deliver in terms of sustainability, health and resilience [25,26,27]. Within the figure, this is reflected in the phrases claiming that biomimicry helps “shift our perspective”, “see design problems and objectives differently”, and better understand “how life works”. Three webpages go beyond sustainability and state that it is time for designs that are not only sustainable, but also “regenerative and restorative – supporting the fabric of life”. One of these websites (BiDL China) uses this notion of restoration and positivity to distinguish biomimicry from the “negative and destructive” goals of biomimetics.

#### 3.2.3. Transformation

Owing to the expressed ideals of increased health, redesign, and reformation, biomimicry, as it is communicated by the BGN, appears to supersede aims of innovation and sustainability by advocating for transformation. The text analysis suggests that the way biomimicry is talked about rejects the culture–nature dualism as defined by domination and extraction, and instead is more aligned with enhancement and education. Within the text this is evident through the recurring themes of “fitting in”, “redesigning”, and “making things right”. Biomimicry is advertised as a method to “redesign the human presence on Earth” (Biomimicry Iberia). It is suggested as a creative and innovative tool to use “in the search for a future that does not compromise the wellbeing of the next generations”, as well as a resource that “promotes a more significant and reciprocal human presence with other species” (Biomimicry Mexico).

Much of the text within the corpus is concerned with re-evaluating humanity’s relationship with nature. Variations of the phrase, “biomimicry is a new way of viewing, valuing, and learning from nature” appear throughout the corpus. This perspective links to the use of the word “system”, in which sentiments of recognizing humanity’s role in a living system and thus calling for systemic change are prominent. This notion is repeated nine times throughout the corpus with sentiments such as, “the biggest mistake we’ve made is forgetting that we are part of, and not separate from, the ecosystems of our planet” (Biomimicry Germany) and “in practice [biomimicry] is dedicated to reconnecting people with nature, and aligning human systems with biological systems” (Biomimicry Los Angeles). Biomimicry is said to offer a chance to make things right, to “embrace a systems view of our world and begin living within planetary bounds” (Biomimicry Germany).

Although, most webpages of the BGN have a normative undertone, messages of hope and relief are unique to the webpage of the Biomimicry Institute. Their message is framed around the supposed hopelessness surrounding the current climate crisis. As a relief from this hopelessness, the Institute offers up biomimicry saying, “Biomimicry brings us hope, because we know the solutions for our greatest challenges are here, accessible, and validated by many of the species still alive today.” The tone of this message, and that of the rest of the BGN narrative, instills a fascination with biomimicry as the answer to humanity’s sustainability challenges.

## 4. Discussion

The analysis highlighted that the promises of biomimicry, as presented by the Biomimicry Global Network (BGN), are expressed as potentials in the realms of innovation, sustainability, and transformation. Through consequently adhering to the use of similar words and phrases, definitions, and references, BGN affiliates seem to converge around a shared discursive understanding of both the promises and presuppositions of biomimicry and its place in a larger context—around a shared imaginary of biomimicry’s role in the future. Such “discourse coalitions” are important, because “discourses frame certain problems; that is to say, they distinguish some aspects of a situation rather than others” [28]. As such, the particular storylines and terms that the BGN propose are code for a wider set of assumptions and presuppositions, for a wider imaginary. Here, the discursive use of the categories’ innovation, sustainability, and transformation, are used to discuss and contextualize the collective findings of the analysis. Within the realm of bio-inspiration, innovation and sustainability are seemingly intertwined and are therefore discussed together. The design paradigm of the BGN network, and resultantly that of Janine Benyus, is discussed and contrasted with the work of MIT professor and media ecologist, Neri Oxman, who is affiliated with biomimicry but not explicitly connected to the BGN. Oxman’s work within the MIT media lab, Mediated Matter, presents a biomimetic design paradigm of a different kind and is thus useful in critically analyzing the collective message of the BGN. The transformative potential of biomimicry’s promise is discussed in relation to the change it offers society, and is therefore discussed under the heading: Biomimicry for Society.

### 4.1. Biomimicry for Innovation and Sustainability

Conceptually, “biomimicry”, as defined by Benyus and others whose motivation seems to stem from a desire for a more sustainable human society, has been described as an ecological form of technological innovation [29]. From the text analysis, it is clear that the practice aims to achieve not only innovative technologies, but technologies that could be sustainable and even regenerative (focusing on the engagement of the whole system as defined by Reed [30]) to their surrounding environment. The promise of biomimicry is aligned with the “biomimetic promise” [6] in pursuing sustainability through the emulation of evolutionary optimization. The promise of inherent sustainability, however, has met with opposition [13]. As Blok and Gremmen [29] explain, the loose definition and underdeveloped philosophy of biomimicry raises questions about whether its promises can be fulfilled and under what conditions. They go on to differentiate a strong and weak concept of biomimicry and the respective ability of each to deliver biomimicry’s promises. The concepts of strong and weak biomimicry are differentiated by their perspectives on nature, technology, and ethics (as dimensions of the biomimicry concept). These categories, and the division of strong and weak biomimicry, provide an effective means of assessing the plausibility of biomimicry’s promise as presented by the BGN. Thus, using this approach, the promises expressed in the figure and corresponding review are discussed using various views of nature, relationships to technology, and the ethical approaches of practitioners.

#### 4.1.1. Nature

Before realizing any of the expressed promises above, biomimicry is dependent on understanding principles of nature. Throughout the text corpus it is implied that tapping into nature’s wisdom is endless, free, and easy to come by. This is clear from a sentence that says, “When it comes to innovation, *nature offers 3.8 billion years’ worth* of insights and clever adaptations” (emphasis added). This sentiment is reinforced by notions such as “seeking nature’s advice” or turning to nature as an “expert consultant”. Abstracting natural principles for any type of biomimicry relies on a particular view of nature. The strong concept of biomimicry is characterized by a view of nature as perfect, and fully accessible [29]. This notion aligns with the BGN narrative and is potentially overconfident of humanity’s ability to understand and “know” the intricate functioning of nature’s systems. In order to learn from various natural systems, patterns and principles need to first be revealed by means of advanced skills and technologies [31]. Ultimately, bridging the functional principles from nature to design requires detailed analyses at multiple levels and an understanding of underlying mechanisms [27]. Although the functional patterns that are foundational to biomimetic design are useful “rules of thumb”, they do not render nature as completely intelligible to us [7]. To assert this would claim an exclusive ability and privilege of knowing and representing the natural world [32].

On the other hand, the weak concept of biomimicry suggests that nature is complex, temperamental, and deficient in its conceptualization [29]. It also implies that humanity has the imaginative capacity to make improvements on nature’s constrained designs. To Neri Oxman, the beauty of being inspired by nature is where mimicry ends and editing begins [33]. The webpage of Mediated Matter speaks of “nature-inspired design and design-inspired-nature” [34], emphasizing this relationship with nature. Building upon natural inspiration with human analogical thinking is trademark of the weak concept of biomimicry and of Neri Oxman’s work. The weak concept is said to leave more room for flexibility in integrating nature into concrete biomimetic designs, whereas the strong concept relies on an arguably naïve perception of a perfect and perfectly knowable nature [29]. Both perspectives inform the relationship between biomimicry and technology and whether designs match the transformative promise of the field.

Affiliates of the BGN have expressed promises to move away from an exploitative exploration of nature in expressing revaluation of natural resources. However, the technification of nature may result in a collection of intellectual resources that can be employed to advance techno–industrial progress [23]. To critical observers, this means that what is mimicked by biomimicry, is not nature *an sich*, but rather a decidedly technological view of nature categorized in terms of the functions and capabilities it can offer humanity [20,29,31]. Despite their differences, both the weak and strong interpretation of biomimicry risk compartmentalizing nature for innovation, viewing it as an “immense, outdoor laboratory that can subsequently be emulated by tools and technologies developed in human-made laboratories” [31] (p. 3). This view contradicts the promise of eliminating human supremacy and negates the implied shift towards intrinsically valuing nature.

As a popular example, one can consider the mimicking of a spider’s silk. Using local resources, and without producing toxic waste, the golden orb weaver spider creates a super-fiber which is five times stronger than steel, and 30% more flexible than nylon [23]. Biomimic, Christopher Viney, hopes to comprehend and mimic this spider’s chemical processes to weave a commercial material five times as effective as Kevlar. However, in the emulation process, the spider becomes valued not for what it is, but rather for what it is able to do, and how [23]. In the wild and in the laboratory, the liveliness of this species is “dissected, pulled apart, and reconstituted as an assemblage of capacities” [23] (p. 73).

In this technification of nature, biomimicry might actually perpetuate the techno–social structures it claims to overcome [23,31,32]. The promise of a new, healthier, interconnected era pales against the technification and valorization of nature. If enclosed as a repository full of seemingly endless possibilities for innovation and capital, biomimicry becomes entangled with domination, control, and capitalism [11,23]. As an extension of this thought, nature as intellectual property also runs the risk of being militarized [32]. For Benyus, this is not an idle worry. She states: “We flew like a bird for the first time in 1903, and by 1914, we were dropping bombs from the sky.” She suggests that perhaps in the end, it will not be a change in technology that will bring us to the biomimetic future, but a cultural and ontological shift that allows a more humble relationship with nature [5] (p. 8). This notion is echoed across the literature emphasizing that biomimicry will only achieve its overall goals if it is not pursued purely as a technological endeavor [31]. As is evident in the promises of the BGN, an ethos of sustainability and respect for nature are part and parcel of the biomimicry narrative. At its core, biomimicry is about a particular ethos from which its technology flows.

#### 4.1.2. Technology

The promises of the Biomimicry Global Network imply that biomimicry offers a way to re-design the way humanity inhabits the Earth. The promise explores the integration of human technology within natural ecosystems. The strong and weak concepts of biomimicry suppose either a separation between technology and nature or a supplementing of nature with technology. These relationships can be described using the design paradigms of Benyus and Oxman. The comparison rests on the idea of supplementing biomimicry using existing and emerging technologies.

Benyus’ biomimicry presents a science of nature; discovering and then emulating nature in technological apparatus [32]. This definition aligns Benyus and the BGN with the strong concept of biomimicry that aims to mimic nature in detail, perceiving it as perfect and separating it from technological improvement [29]. If widely adopted, however, biomimicry that only aims to mimic nature could give rise to a wholly artificial world, where pollination is outsourced to robotic bees and carbon is sequestered by artificial trees [20]. Furthermore, what cannot always be mimicked by the strong concept of biomimicry is the adaptability of a natural system to the context and history in which it is embedded. To overcome these risks, Benyus and associated practitioners emphasize the importance of integrating biomimetic designs as best as we can into nature’s dynamic circular systems [5,10,35].

Oxman’s biomimicry, on the other hand can be considered as a technology of nature: her work shifts from mimicking nature to editing and integrating it within design [32]. This type of biomimicry is not bound by a reference to the original model and is thus able to create something entirely new [32]. Oxman’s work aligns with the weak concept of biomimicry, as it both supplements nature with technology and views nature as a deficient but improvable model [29]. In such a way, the weak concept of biomimicry suggests a way to create a new context and history for a biomimetic technology, ensuring it is well-integrated into its surroundings [29]. These considerations are vital for influencing biomimicry’s promise of humanity fitting within nature, and potentially offering a way to “make things right”.

#### 4.1.3. Ethics

Biomimicry, as defined by the BGN, involves learning from nature not only in terms of how to design, but also how to design appropriately within the biosphere. In presenting it as such, biomimicry aims to establish an ethical standard to assess the designs it creates [5,20]. In this framework, nature is conceptualized as a normative principle on which to gauge the appropriateness, ecological health, and integrity of biomimetic designs [29].

The strong concept of biomimicry employs this ecological standard to evaluate biomimetic designs. Using nature as a measure in this regard is to risk committing the so-called naturalistic fallacy—arguing that something is good because it is natural [29]. In suggesting that biomimetic designs are good, purely because they are based on natural principles, the strong concept of biomimicry becomes vulnerable to committing this fallacy [20]. Furthermore, it must be acknowledged that in mimicking nature, there is a necessity to translate and interpret natural phenomena. Within the strong concept of biomimicry, this necessity challenges the ethical advantage that it uses to claim sustainability. In translating a natural phenomenon, the intention moves from discovery to transposition and invention, thus undoing the naturalistic ethic it claims to hold.

Although learning from living nature promises extraordinary quality of technical applications, nature is not always sustainable in the sense of human-made visions [2]. Therefore, biomimetic designs ought to be sustainable, but should undergo similar sustainability assessments of conventional designs. Mead and Jeanrenaud [9] assess the connection between biomimicry and measures of sustainability in their paper *The Elephant in the Room: Biomimetics and Sustainability*. From an exploratory survey questioning practitioners of biomimicry, they found no established system of accountability when it comes to biomimicry and sustainability [9]. Furthermore, although a variety of sustainability metrics were employed, many practitioners relied on an intuitive (rather than science-based) evaluation of their designs [9] (p. 7). Mead and Jeanrenaud [9] thus called for a more formal dialogue regarding sustainability as well as for science-based ecological targets.

In the context of innovation and sustainability, both the strong and weak concepts of biomimicry pose some risks. Blok and Gremmen [29] compared the two strengths and concluded that the strong concept of biomimicry was more problematic than the weak, based on its naivety and vulnerabilities. However, the authors were hesitant in advocating for a full shift toward the weak concept of biomimicry, given the “responsible and ecological” underpinning of the strong concept [29] (p. 216). Both concepts can be used to further technological innovation, but perhaps it is their ethical standpoint, and management of potential risks that positions them on the scale of sustainability.

The creation of sustainable designs does not necessarily depend upon a particular design methodology, but rather on cultural and societal norms and values [10,22,31,36]. This is a continuation of ethics beyond judging the rightness of designs, towards reflecting on the underlying intentions for practicing biomimicry. The true hope of the strong concept of biomimicry and that of the BGN is expressed by Benyus when she states: “At its best biomimicry should take us aback, make us more humble, and put us in the learner’s chair, seeking to discover and emulate instead of invent” [5]. She stipulates further that, if we are to use the tools and technology we have, with this underlying motivation, it holds that our basic relationship with nature, as well as the story we tell ourselves about who we are in this universe, has to change [5,7]. The BGN narrative firmly aligns with the strong concept of biomimicry. Table 2 summarizes the characteristics of the strong and weak concepts of biomimicry and highlights the contrasting outcomes of biomimetic design paradigms. Arguably, the different ethical approaches to each concept will have a significant effect on whether biomimicry contributes to innovation in an exploitative or sustainable manner.

### 4.2. Biomimicry for Society

Biomimicry views and presents itself as a revaluation of design and technology. The promised transformational potential of biomimicry is evident throughout the analysis as expressions of empathy, hope, and relief, as well as a shifting relationship with nature. Biomimicry advocates suggest that biomimicry is not merely a technique, but could define a “novel and ethical political, economic, and social order” [32] (p. 797). Rather than a simple technological innovation then, biomimicry presents a new discourse on the relationship between humans, technology, and nature—alternative storylines of how to be in the world. In its practice, which is more active than reflective, biomimicry aims to establish a relationship with nature and an appropriate social ethic [5,32]. This message seems to stem from the Biomimicry Institute, but also underlies the entire BGN narrative.

Biomimicry relies on a bio-inclusive ethic; one that assigns value to all species, enabling them to fulfill their purpose out of respect and consideration [7]. This ethic links biomimicry to the deep ecology philosophy that insists all organisms and living systems have equal right to exist and thrive on the planet, not because of their potential value to humanity, but simply because they exist [7,24]. Given this inherent value, a notable desire expressed in the BGN narrative is to better understand and integrate more effectively into natural ecosystems. Biomimicry adheres to this value in its application of ecological knowledge to the human endeavors of urban design, architecture and engineering [24].

The underlying philosophy can be traced back to the idea of sustainable participation in nature as opposed to the control of it. This notion implies that humanity is part of and dependent on the health of ecosystems and the health of the biosphere [10], which is a clear sentiment expressed in the findings. In the current era, the complex interactions between social, cultural, economic, and ecological systems are being acknowledged and perhaps slowly, an ecologically and socially literate worldview is emerging [10]. What is suggested as a subsequent requirement is a design approach that is synergistic in increasing human, societal, and ecological health. Within this mode, agents (or participants) must pursue their goals in ways that use, but do not disturb, the integrity of others, whilst at the same time proving to be reciprocal [7]. To achieve these conditions in a particular place and facilitate system-wide health, designers will have to shift from the detached view of culture as apart from nature to a more holistic and participatory perspective of culture as part of nature [10]. In this sense, biomimicry should be a design approach that is diverse, transdisciplinary, and participatory. To fully act from within nature as the BGN and biomimicry advocates have suggested, more is needed than new technological inspiration. Rather, it needs a cultural, epistemological, and ontological shift, redirecting current ways of doing and being to more sustainable directions.

## 5. Conclusions

This research has highlighted how the promise of biomimicry, as expressed by the Biomimicry Global Network, takes a different shape concerning innovation, sustainability, and society. The outcome has proven informative and intricately woven with the intentions of design. The innovative potential of biomimicry has been shown to require some technical considerations but is most easily realized, whereas promises of sustainability and transformation require more fundamental shifts in levels of emulation and in how humanity values nature.

If one were to imagine the future according to proponents of biomimicry, it is clear from the assemblage of the data and literature examined, that the vision requires significant consideration. What has been shown, is that the outcome of a biomimetic design changes significantly depending on whether nature is viewed as a technological entity for exploitation; deemed perfect; acknowledged as a complex, deficient system; or considered as part of and not separate from culture. Thus, the imaginaries and discourses of biomimicry matter. Practicing biomimicry with these various perspectives could create different future imaginaries. Although both guided by the promises of biomimicry, the future according to Benyus differs significantly from the future according to Oxman. Still, both visions for the future might catalyze meaningful changes towards a greater sustainability of human society [37].

Design at its root, is the expression of intentionality through interaction [10]. An overarching finding from this analysis, is that intentionality and cultural norms are crucial elements to consider when practicing biomimicry; arguably the most important. Rather than yet another hubristic search for the technological sublime, biomimicry’s ethics and practice should rely on “technologies of humility” [38] and forms of “institutionalized reflexivity” [39], in which a more humble reflection on the effects of the technology are encouraged. For a practitioner or an aspiring one, it is arguably essential to reflect on how different intentions within the field could result in largely contrasting outcomes, or even ones that contradict the initial promise [25]. Ultimately, as humanity continues to investigate and understand the concepts of life, careful consideration should extend towards creating what is possible but harmful, and what is possible and *useful* [36].

In the deepest sense, biomimicry presents a re-evaluation of humanity’s desires and values [7]. It requires a transition to mutualistic engagement with nature, and a shift in thinking about ways to achieve consumer wants and needs within the interests of nature, meaning a rethinking of consumer demands entirely. Adopting such an approach requires a deep understanding of the intricacies of life systems and communication with nature [22]. Although biomimicry for sustainability, and certainly biomimicry for innovation could be achieved; without a shift in desires and intentions, biomimicry’s promise of reconnection and a human sense of being a part of, not apart from, the living world might not be fully realized [22].

## Figures and Tables

**Figure 1 biomimetics-05-00033-f001:**
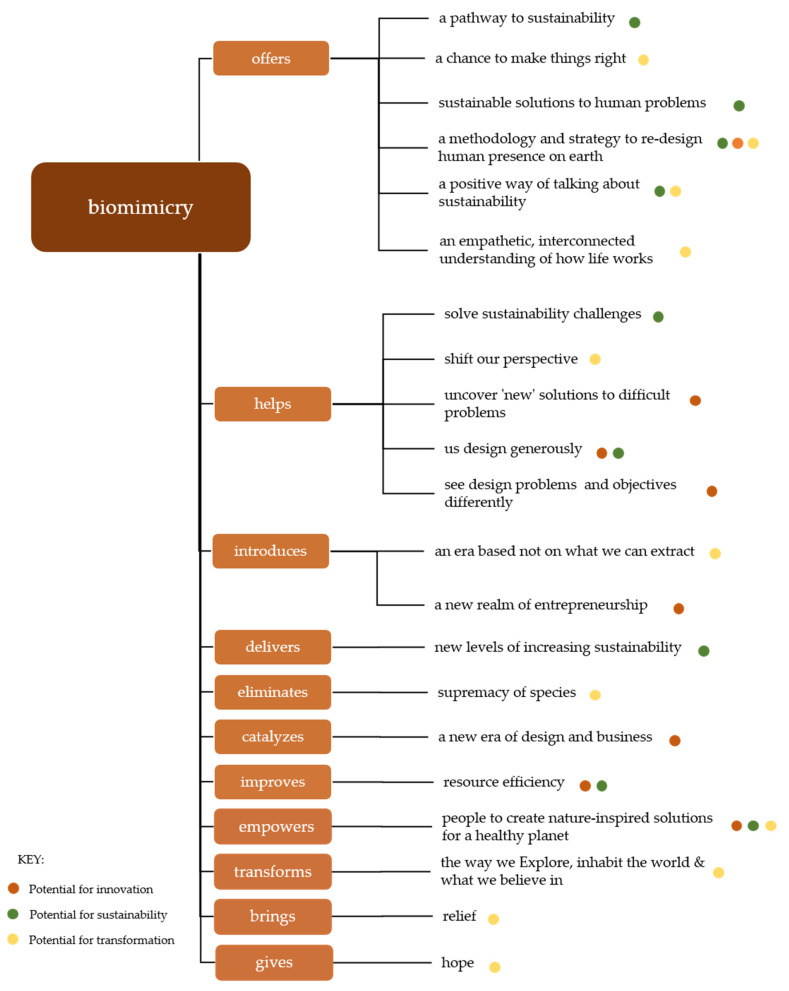
The figure depicts a word tree of quotes from the Biomimicry Global Network.

**Table 1 biomimetics-05-00033-t001:** The webpages of the Biomimicry Global Network used as sources for the review, along with their associated web addresses and content pathways.

Network	Web Addresses	Content
BiDL (China)	bidl.tongji.edu.cn	Homepage
Biomimicry Australia	biomimicryaustralia.org	Biomimicry Consulting
Biomimicry Belgium	biomimicrybe.org	What is Biomimicry?
Biomimicry Chicago	biomimicrychicago.net	Resources >> Biomimicry 101
Biomimicry Colorado	biomimicrycolorado.com	Biomimicry Basics
Biomimicry Europa	biomimicry.eu	Biomimicry?
Biomimicry Germany	biomimicrygermany.com	Updated post research
Biomimicry Iberia	biomimicryiberia.com	Biomimicry
Biomimicry Institute	biomimicry.org	What is Biomimicry?
Biomimicry Los Angeles	biomimicryla.org	About >> Biomimicry
Biomimicry Mexico	Biomimicrymex.org	Bio-qué? *
Biomimicry Netherlands	bioimimicrynl.org	The Biomimicry Story
Biomimicry New York City	biomimicrynyc.com	Biomimicry
Biomimicry Oregon	biomimicryoregon.org	Homepage
Biomimicry San Diego	biomimicrysandiego.org	What is Biomimicry?
Biomimicry Singapore	biomimicrysingapore.net	What is Biomimicry?
Biomimicry South Africa	biomimicrysa.co.za	What is Biomimicry?
Biomimicry Switzerland	biomimicryswitzerland.org	Biomimicry Thinking
Great Lakes Biomimicry	glbiomimicry.org	About >> What is Biomimicry?

* Translated using Google Translate.

**Table 2 biomimetics-05-00033-t002:** Overview of strong and weak concepts of biomimicry following Blok and Gremmen [29] and adapted to include ideals from Fisch [32].

	Strong Biomimicry	Weak Biomimicry
**Practitioner**	Janine Benyus	Neri Oxman
**Mimesis**	Naïve conceptualization of mimicry	Acknowledges supplementarity of mimicry
**Nature**	Presupposes perfection of nature	Presupposes complex deficiency of nature
**Technology**	Separates nature and technology	Integrates and edits nature with technology
**Ethics**	Vulnerable to the naturalistic fallacy	Avoids naturalistic fallacy
**Main risk**	Enclosure and valorization of nature	Exploitation of nature

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
