# Peer review of "Promises and Presuppositions of Biomimicry"

_biomimetics, 2020, doi:10.3390/biomimetics5030033_

Round 1
Reviewer 1 Report
General:
- This article addresses interesting and important concepts for the field of biomimicry. It is well considered, well written and thought-provoking.
- My main concern/comment is that the analysis and discussion do not link clearly enough to the methodology adopted and results generated. There is an opportunity to more clearly highlight how the analysis and discussion were drawn from/ informed by the methodology and results.
Methods and results:
- The methods adopted seem well-suited to the intent of exploring lexicon in the biomimicry space, however I think that the paper would benefit from a clearer articulation of the criteria, method and analysis approaches adopted. This includes:
- Approach to creation of the word tree
- Clarification of the interdisciplinary framework used for assessment (Line 73: “these proclaimed promises were assessed using an interdisciplinary framework drawing on publications in philosophy, social science, environmental anthropology, sustainability and design.”)
- Criteria adopted for determining ‘accessibility, usability and language’ of websites (Line 81: “The webpages used in this study were selected from this map based on their accessibility, usability and language.”)
- You note that coding categories related to lexicon, phrasing and tone, though without additional detail this appears quite subjective/vague. Further insight into the analysis approach would benefit here. Also, was the coding approach peer/co-author -reviewed to manage subjectivity?
- Regarding content drawn from the web pages, were there any controls to manage the amount of text copied from each? Note how this was managed to confirm that variations in text quantity did not materially impact the findings of the text analysis.
- Line 104-111 – is this finding expanded on or discussed elsewhere in the paper?
- How were the categories of innovation, sustainability and transformation selected?
Discussion:
- The exploration of ‘nature’ and weak and strong biomimicry in the discussion is well articulated, considered and important.
- The discussion is well written and with interesting insights. It would benefit from being more clearly linked back to the research findings, to ensure that it does not appear as simply an extension of the literature review.
- You may be interested in ‘The dialectical environment of the mind’ by Robert R Windle III, which explores philosophical foundations of biomimicry
Specific minor comments:
Line 31: Remove ‘the’ before Benyus
Line 59 – suggest clarifying the term ‘biomimic’ before first use
Line 219-220 – ‘More sources’
Line 266: Remove open parentheses.
Line 270: remove additional space.
Line 336: Citation error (‘and Mathews’)
Reviewer 2 Report
Please see attachment for comments.

Reviewer 3 Report
This is a very nuanced and comprehensive analysis of the promises and presuppositions of biomimetics. Text mining techniques are used to analyses the discourse, resulting in a word tree, but this is supported by an in-depth conceptual analysis of paradoxes and challenges of the biomimetics promise. The discourse aims to move beyond nature-culture dualism, trying to redesign design-thinking to become more sensitive to the natural environment, learning from rather than exploiting nature. We seek nature’s advice while nature is not necessarily perfect nor perfectly knowable. Two positions (strong and weak biomimicry) are subjected to a comparative analysis, resulting in the conclusion that basic prepositions and images concerning nature must be taken into consideration given their practical implications for the actual development and objectives of biomimicry, as both positions outline diverging scenarios for the future. As a comprehensive review of the literature, demonstrating the impact of guiding conceptions, this is a clear, relevant, balanced and helpful contribution to biomimicry discourse as well as to biomimicry as an evolving practice.
Round 2
Reviewer 2 Report
The paper is very much improved. Well done!
-Though the introduction is much more robust than it was before, there is still no literature review in the paper.
Line 65 -"the hope is that as observers,..." Avoid colloquialisms.
Lines 70-74 - Repetitive sentence structure with "Promises matter..." and Promises and expectations matter..."
Line 76-82 - "Not only do promises structure decision-making in the present, they also co-produce an imaginary about what a desirable future ought to be like. Such socio-technical imaginaries, collectively held, institutionally stabilized, and publicly performed visions of desirable futures, animated by shared understandings of forms of social life, and social order attainable through, and supportive of, advances in science and technology, then further come to shape technological development, impacting the kind of future worlds that are possible [19]". This is a very powerful framing of futurism and well-placed in the text. However, the sentence is cumbersome and distracts from the message. Consider revising into two sentences.
Line 92: ..."an investigation into the imaginary of biomimicry as proposed by the BGN..." Imaginary or imagery? Check usage throughout. Either could work, imaginary is just evocative of something different than imagery.
Line 107 - "of the practice; revealing its". Reconsider semi-colon use.
Line 157 -"text analyses acted as..." Analysis?
Line 210 - "Separate" or distinguish?
Line 280 - "The design paradigm of the BGN network, and resultantly that of Janine Benyus, is discussed and contrasted with the work of MIT professor and media ecologist, Neri Oxman, who is affiliated with biomimicry but not explicitly connected to the BGN." "affiliated" implies a formal relationship. Perhaps "practices biomimicry but is not explicitly connected...".
Line 292: ..."The promise of biomimicry is this aligned with the ‘biomimetic promise’..." ???
Line 301-303 - "Thus, using this approach, the promises expressed in the figure and corresponding review are discussed using various views of nature; relationships to technology, and the ethical approaches of practitioners." Confusing sentence structure.
Line 351: "might actually perpetuate" - Avoid colloquialisms i.e., actually.
LIne 370 - "of Benyus and Oxman,." typo
Line 383: "Oxman’s biomimicry, on the other hand can be considered". Add additional comma.